# Compositional Data and Microbiota Analysis: Imagination and Reality

**DOI:** 10.3390/microorganisms12071484

**Published:** 2024-07-20

**Authors:** Tatsuki Itagaki, Hirokazu Kobayashi, Ken-ichiro Sakata, Ikuya Miyamoto, Akira Hasebe, Yoshimasa Kitagawa

**Affiliations:** 1Oral Diagnosis and Medicine, Faculty of Dental Medicine, Graduate School of Dental Medicine, Hokkaido University, Kita-13 Nishi-7, Kita-ku, Sapporo 060-8586, Japan; titagaki@den.hokudai.ac.jp (T.I.); h.kobayashi@den.hokudai.ac.jp (H.K.); sakata-0303@den.hokudai.ac.jp (K.-i.S.); ikuyam@den.hokudai.ac.jp (I.M.); ykitagaw@den.hokudai.ac.jp (Y.K.); 2Microbiology, Faculty of Dental Medicine, Graduate School of Dental Medicine, Hokkaido University, Kita-13 Nishi-7, Kita-ku, Sapporo 060-8586, Japan

**Keywords:** microbiome, compositional data, ratio analysis

## Abstract

The relationships among bacterial flora, diseases, and diet have been described by many authors. An operational taxonomic units (OTUs) are the result of clustering the 16S rRNA gene sequences at a certain cutoff value, and they are considered compositional data. As Pearson’s correlation coefficient is difficult to interpret, Aitchison’s ratio analysis was used to develop a method to handle compositional data. Multivariate analysis was developed because univariate analysis can be subject to large biases. Simulations regarding absolute abundance based on certain assumptions and some analyses, such as nonparametric multidimensional scaling (NMDS), principal component analysis (PCA), and ratio analysis, were conducted in this study. The same content as a 100% stacked bar graph could be expressed in low dimensions using PCA. However, the relative diversity was not reproducible with NMDS. Various assumptions were made regarding absolute abundance based on the relative abundance. However, which assumptions are true could not be determined. In summary, ratio analysis and PCA are useful for analyzing compositional data and the gut microbiota.

## 1. Introduction

Microbiome analysis is a field in microbiology and information engineering [1,2]. In recent years, the relationships among bacterial flora, diseases, and diet have been described. Operational taxonomic units (OTUs) or amplicon sequence variants (ASVs) are the result of clustering 16S rRNA gene sequences at a certain cutoff value [1]. Various methods can be used to normalize the data. With total sum scaling (TSS), a composition is created with a predetermined number of reads [1] and, sometimes, by determining the lead depth based on a certain index.

The basis consists of non-constrained data [1,3,4,5,6,7], whereas compositional data indicate relative information using positive values between 0 or 1 that do not include 0 or 1 [1,3,4,5,6,7], which are often visualized using 100% stacked bar graphs. Pearson’s discussion of correlation coefficients for compositional data in 1897 marked the beginning of the analysis of compositional data [3]. The analysis of compositional data did not progress for another 89 years after this. An innovative approach proposed by Aitchison (1986) and many successive authors have enabled this difficult task to be completed by mapping the simplex space to the Euclidean space or by building a proper algebraic function within the simplex (stay-in-the-simplex method) [4,5,6]. In 2011, Ohta et al. identified certain components whose absolute abundance remained unchanged using only compositional data [7]. Using Ohta et al.’s method, changes in absolute abundance could be estimated using relative abundance data [7].

Univariate analyses, such as the chi-squared test, were developed, followed by the development of multivariate analyses to reduce bias [1]. In the early 2000s, further developments in the analysis of compositional data were achieved. In the field of microbiology, nonparametric multidimensional scaling (NMDS) and linear discriminant analysis effect size (LEfSe) became popular with advances in computing. In the field of geosciences, principal component analysis (PCA) is a mainstream method [1]. LEfSe is a univariate analysis method, whereas PCA is a multivariate analysis method. NMDS and PCA handle data differently [1]. NMDS and PCA cannot be used for the same kind of data [1]. It is common to apply PCA to the compositional data [1].

## 2. Compositional Data and Microbiota Analysis

OTUs and ASVs are compositional data [1] which indicate relative abundance [1,2,8]. The same composition can be obtained via random sampling [1,3,4,5,6,7], whereas unbalanced sampling results in different compositions [1,3,4,5,6,7]. OTUs and ASVs do not indicate absolute abundance [1]. In other words, absolute abundance likely varies from sample to sample [1]. TSS is commonly used in many fields [1,4,5,6,7], being a method that weights data based on the abundance within a single sample. TSS does not originally contain zeroes, but zeroes are sometimes used for convenience [1]. When comparing samples, differences in the samples can be identified if there are zeroes [1]. However, no further comparison is possible because the apparent abundance changes if the constituent components differ [1,3,4,5,6,7]. The absolute abundance may be low when comparing samples with the same constituents, even if the occupancy is high. This is because if the absolute abundance is low, the relative abundance appears high; however, the opposite may also be the case.

## 3. Discussion

### 3.1. Artificial Data and Statistics

Using artificial data can demonstrate imaginary or real results in microbiome analysis. Assuming six types of bacteria A to F, their relative abundances are as shown in Appendix A. OTU data usually only reveal relative abundance. Assumption 1 is shown in Appendix A, where the total absolute bacterial abundances are almost the same. Assumption 2 is the case where the total bacterial counts are different, as shown in Appendix A. In Appendix A, the compositions of the microbiota among samples are the same (or nearly the same), although the absolute abundances are different. The statistics in Appendix A are shown in Table 1.

The values of the statistics between compositional data and Assumption 1 are close because neither of them weighs the reliability of the data. However, the statistics of Assumption 2 are different from those of the compositional data because Assumption 2 is weighted based on the reliability of the data. Many studies on microbiome analysis have adopted Assumption 1 [1,2,8]. For Assumption 1 to hold, the total absolute abundance of the bacteria must be shown to be the same among individuals or over time [1]. Assumption 2 is probably more realistic because the total absolute abundance of the bacteria is unlikely to be the same among individuals or over time [1]. However, obtaining weighted data for Assumption 2 from relative abundance data is impossible [1].

### 3.2. NMDS and PCA

The misconception about β-diversity is as follows [1]: Many researchers have used the calculation results from NMDS as the β-diversity [1]; however, NMDS cannot be applied to compositional data [1]. Figure 1 shows an example. All the data (Appendix A) were analyzed using the Vegan 2.6-4 package in R version 4.3.1 (2023-06-16 ucrt). The NMDS calculation results do not make mathematical sense [1]. The plots were not reproducible when calculated using the Jaccard distance. Despite the low explanation rate of the principal coordinates in NMDS, the common misconception is that each principal coordinate has meaning [1]. Moreover, ANORSIM, ADONIS, and other analyses may result in small *p*-values and small R and R^2^ values [1], which only indicate a lack of consistency among the numerical data, statistical model, and null hypothesis, probably resulting from the correlations among the bacteria being ignored [1]. Although the PCA results are produced on different scales (Figure 1), the unweighted compositional data and Assumption 1 match the PCA results [1]. Different results were obtained with Assumption 2 and the compositional data, which have different data weights. Although data weighting was ignored, PCA could still describe the relative abundance [1]. In other words, the same content as a 100% stacked bar graph could be expressed in two dimensions using PCA [1], demonstrating the difficulty in analyzing relative abundances.

### 3.3. LEfSe

The LEfSe algorithm repeats the univariate analysis and detects significant differences. Repeating the univariate analysis increases the probability of false positives. Many people have wrongly assumed a significant difference that is detected mathematically to be a scientifically meaningful difference. The mathematical difference is merely a necessary condition, so we should ensure that the difference is scientifically meaningful. The LEfSe algorithm includes the Kruskal–Wallis test, a nonparametric test that uses ranked data within the whole. However, compositional data only show the rank within the sample [1]. Compositional data cannot be ranked in absolute abundance among samples [1]. In other words, if other samples are included, the overall rank cannot be determined, so the Kruskal–Wallis test cannot be used. Moreover, the correlation coefficient is used in the LEfSe algorithm. However, Pearson proved that the correlation coefficient is different from the original correlation coefficient for compositional data [1,3]. This means that LEfSe has no scientific value because it repeats tests that cannot be used mathematically. However, one of the LEfSe prerequisites is the strict condition that the rankings of the relative abundance and absolute abundance must match. If this prerequisite is met, the OUTs data can become count data, and a different multivariate analysis can be applied. LEfSe is difficult to be used to discover something scientifically meaningful. LEfSe provides useless results unless evidence can be provided that the rankings of relative abundance and absolute abundance match.

### 3.4. Ratio Analysis and Method of Ohta et al.

The analysis of compositional data was systematized by Aitchison [1,4,5,6,7]. The comparison of compositional data is only possible between data of the same constituent elements [1,3,4,5,6,7]. As such, ratio analysis is commonly used to analyze compositional data [1,4,5,6,7]; however, the results of ratio analysis widely vary depending on which component is used to calculate the ratio [1,4,5,6,7,8]. When considering a ratio, components whose absolute abundance remains unchanged are the best to use [1,7]. From relative abundance data, increases and decreases in the absolute abundance of other components can be estimated based on the components whose absolute abundance remains unchanged [1,7]. However, identifying components whose absolute abundances do not change based on relative abundance data is difficult, and constant components may even be lacking [1,7]. The desired components for selecting the ratio can be estimated [1,7]. Ohta et al.’s method may be helpful in determining which component should be used as the denominator in ratio analysis [1,7], in which the coefficients of variation for pairwise ratios are compared, and the numbers of components for which the coefficient of variation increases are counted [7]. The coefficient of variation in the compositional ratio is subject to change when the unchanging component is switched between the denominator and numerator, and the coefficient of variation tends to be small when the unchanging component occurs as the denominator against any arbitrary component (Test 1) [7]. The coefficients of variation for all the ratio combinations are compared [7], and the four or five lowest coefficients of variation are selected [7]. The ratio of the component pair that yields the lowest coefficient of variation is most likely to represent the two unchanging components (Test 2) [7]. The component that repeatedly appears in these four or five ratios is the candidate as the unchanging component [7]. However, Tests 1 and 2 are not a necessary and sufficient condition for uniquely identifying the unchanging components [7]. The R script is available in the Japanese publication by Ohta.

Data from repeated measurements can be used to predict increases or decreases in the bacterial abundances within an individual [1]. For Assumptions 1 and 2, the most invariant component from the coefficient of variation is B. In other words, if Assumptions 1 or 2 are correct, the increase or decrease in the absolute abundance of other components can be estimated by using the ratio of B in the compositional data. Identifying B as the component with the most constant absolute abundance is difficult based on compositional data alone. Test 2 in the method of Ohta et al. detects that B is the most likely to be unchanged.

The Firmicutes/Bacteroidetes ratio is often used in microbiome analysis [1,8]. The abundances of Firmicutes and Bacteroidetes can be in three possible states—increasing, unchanging, or decreasing. Specifically, the following nine patterns are possible: increase/increase, increase/unchanged, in-crease/decrease, unchanged/increase, unchanged/unchanged, unchanged/decrease, de-crease/increase, decrease/unchanged, and decrease/decrease. As such, the F/B ratio can be increasing, unchanging, or decreasing. The F/B ratio might change depending on the increase or decrease in Firmicutes abundance if the Bacteroidetes abundance is unchanged. The main advantage of the Ohta et al. method is allowing the estimation of the changes in the microbiota within individuals [7]. Therefore, the changes in the absolute abundance of bacteria within individuals can be estimated if the unchanged components are known. This method can be used to standardize (estimate the absolute changes in the basis data) 16S rRNA data, so that relative abundances and problems with compositional data may be overcome. Studies should provide a reason for using the F/B ratio but, to date, none have. A relative change is a change that is apparent but differs from the actual amount of change [1,3]. The F/B ratio has little importance if it reflects relative fluctuations. Furthermore, the study results based on incorrect assumptions have been reported. In the field of microbiota analysis, a study that conducted NMDS had no scientific basis [1]. In the field of geosciences, errors in analysis have been acknowledged and corrected [4,5,6,7]. As such, the time has come to admit and correct mistakes in the field of microbiota analysis [1,2]. Relative variation can be expressed using principal component analysis [1]. Ohta et al. provided a method (the coefficient of variation method) to estimate the absolute changes in the basis data using compositional data [7], which is valuable because the actual change rather than apparent change is estimated [7]. Using PCA and ratio analysis with the Ohta et al. method provides a novel strategy in the study of the human gut microbiota. If the absolute abundance of bacteria in each individual and the relationship between each bacterium and the number of OTUs are known, we would be completely free of compositional data.

## 4. Conclusions

OTUs indicate relative abundance. Compositional data are standardized data within a sample. Relative abundance can be analyzed using PCA or ratio analysis, meaning that they can be used to study relative (apparent) changes. Various assumptions can be made regarding absolute abundance based on relative abundance. However, determining which assumptions are true is impossible. Combining microbiota analysis with other methods or technologies is necessary to make comprehensive judgments due to the limited amount of information that can be learned from relative abundance.

## Figures and Tables

**Figure 1 microorganisms-12-01484-f001:**
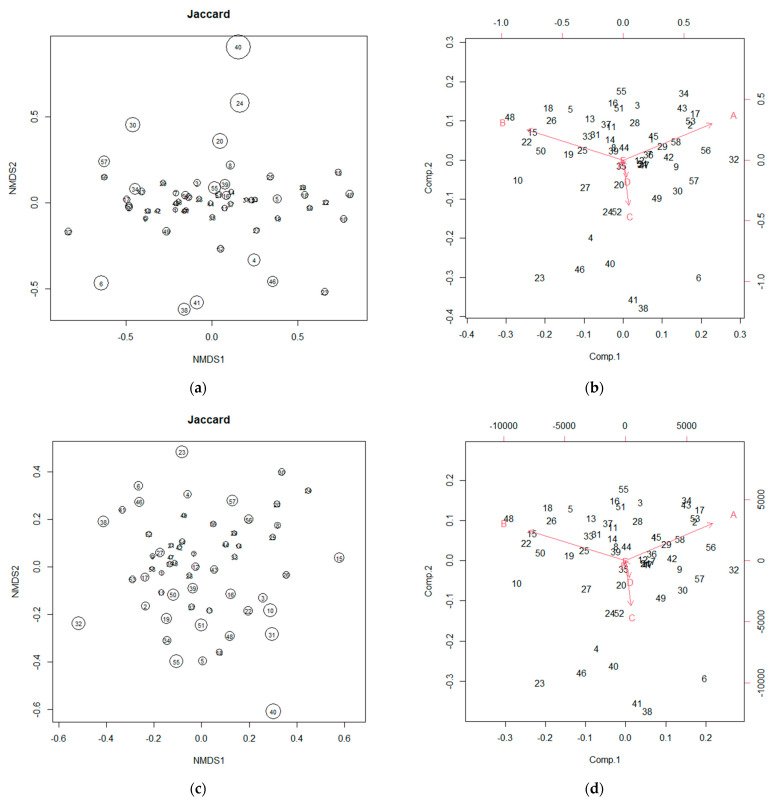
Relative abundance: (**a**) NMDS of compositional data. (**b**) Biplot of compositional data. (**c**) NMDS with Assumption 1. (**d**) Biplot with Assumption 1. (**e**) NMDS with Assumption 2. (**f**) Biplot with Assumption 2. Figure 1 shows the change in appearance. The numbers indicate ID numbers.

**Table 1 microorganisms-12-01484-t001:** Statistics of the sample data.

**Composition**	**A**	**B**	**C**	**D**	**E**	**F**
Average	0.394288	0.456073	0.071928	0.051586	0.025462	0.000663
Coefficient of variation	0.333277	0.302911	1.02974	1.149273	2.025749	1.371844
Skewness	−0.14103	0.282245	1.983293	4.330247	3.094253	3.004872
Kurtosis	−0.51101	−0.3021	3.81018	24.34699	10.19621	10.68641
Correlation with A	1	−0.74465	−0.16093	−0.1366	−0.16563	0.029699
**Assumption 1**	**A**	**B**	**C**	**D**	**E**	**F**
Average	3918.19	4541.655	713.4828	513.5	254.2759	6.568966
Coefficient of variation	0.329642	0.306276	1.029157	1.153159	2.028733	1.358041
Skewness	−0.2025	0.290015	2.00465	4.354737	3.095241	2.975476
Kurtosis	−0.58333	−0.30179	3.923947	24.55164	10.19947	10.65289
Correlation with A	1	−0.73827	−0.17343	−0.13419	−0.16025	−0.00122
**Assumption 2**	**A**	**B**	**C**	**D**	**E**	**F**
Average	18411.97	21386.98	2926.345	2638.897	968.5172	29.37931
Coefficient of variation	0.990369	0.965048	1.15911	1.988405	2.365539	1.565635
Skewness	2.002331	1.818849	1.860411	5.413138	4.758781	2.722131
Kurtosis	5.321145	4.519852	3.731474	33.98872	27.75528	8.969854
Correlation with A	1	0.778904	0.555044	0.339023	0.111192	0.320866

## Data Availability

Data are contained within the article and Appendix A.

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
