# Peer review of "Compositional Data and Microbiota Analysis: Imagination and Reality"

_microorganisms, 2024, doi:10.3390/microorganisms12071484_

Round 1

Reviewer 1 Report (Previous Reviewer 2)

Comments and Suggestions for Authors

In this manuscript, the authors pointed out PCA and ratio analysis should be used for compositional microbiota data analysis. The revised version is much improved over the original submission. The authors addressed all my questions and made corresponding changes to the manuscript, which makes it is easy to follow the authors' opinion now.

Author Response

Dear Reviewer 1,

In this manuscript, the authors pointed out PCA and ratio analysis should be used for compositional microbiota data analysis. The revised version is much improved over the original submission. The authors addressed all my questions and made corresponding changes to the manuscript, which makes it is easy to follow the authors' opinion now.

Answer
Thank you very much for reviewing our manuscript and offering valuable advice. You kindly provided valuable comments. It may be difficult to read, but we have used the English Editing MDPI service and revised it to make it easier to read because we want our manuscript to be read by many readers.

Again, thank you for reading the ambiguous and difficult-to-read text and pointing out the appropriate points. Thanks to you, our manuscript has improved. We have worked hard to incorporate your feedback and hope that these revisions persuade you to accept our submission.
Thank you.

Reviewer 2 Report (New Reviewer)

Comments and Suggestions for Authors

This is an interesting paper that describes the usefulness of PCA in analyzing microbiome data. I think it is worth publishing, but the paper itself is not easy to read. The methodology is sound, but the paper itself has to be modified to improve the language structure and style of writing.

For example, this sentence in the abstract: "Simulations regarding absolute abundance based 16 on certain assumptions and some analysis were conducted in this study". What does "some analysis" refer to?

Additionally, the use of the word "fantasy" is not ideal. I urge the authors to use some other word because the word "fantasy" implies not real or in the imagination.

In section 3.1, there is reference made to the supplementary tables. If you are making extensive reference to the supplementary tables, then you may want to consider putting them in the main text.

Figure 1 is difficult to interpret due to lack of color. Is there any way to colorize the figure to point out differences. Also, the legend is too sparse and more information needs to be provided.

Furthermore, the data availability should be different than reasonable request. The data should be uploaded to a site so that other researchers have access to it. In this day and age, "reasonable request" is not a reliable way of disseminating data. All R scripts and any code should be uploaded to Github.

Comments on the Quality of English Language

This is an interesting paper that has to be re-written and re-structured to make sense. It would make sense to have the paper be edited by a language editing service. 

Author Response

This manuscript is a resubmission of an earlier submission. The following is a list of the peer review reports and author responses from that submission.

Round 1

Reviewer 1 Report

Comments and Suggestions for Authors

The authors studied the compositional data and microbiota analysis with focus on the fantasy and reality. and worked on different methods which applies often in microbiota studies. Nevertheless, it is not clear what is the aim and study method/approach, our main aim of the study. The manuscript, including the abstract, and tables (many times, please check if it is complete in the final version), needs extensive polish. 

Reviewer 2 Report

Comments and Suggestions for Authors

In this article, Itagaki et al. tried to express  their opion on compositional microbiota data analysis. After two rounds of careful reading, I figured out that they tried to point out many published microbiota data analysis are not valid because the assumption uderlying the statistical methods used for data analysis is not correct. They further suggested ratio analysis is useful for compositional data analysis if  Ohta et al.'s method is used to find some unchanging component across samples. However, because the English writing is not coherent, it is really hard to figure out the authors' actual opion for microbiota  compositional data analysis. 

Major issues:

1. The authors should overhaul the writing of the article to make easy to understand and express their opinion clearly and explicitly. The current conclusing section is not insightful. Too many short sentences were used and hard to see the connection between sentences.

2. What if no unchanging component can be found for a microbiota dataset?  Do the authors have any suggestion for this scenario?

Minor issue:

1. Tables 1-3 are better be included as supplementary data.

2. Tables 2 and 3, the compositions of microbiota between samples are the same (or nearly the same), though the absolute abundances are different. Please clearly state this fact in lines 70-72.

3. Line 107, "PCA can show relative abundance [1]". Based on my knowledge, PCA is usually used to demonstrate sample variation and dimension reduction. Please clarify, how can PCA show relative abundance?

4. Lines 113-122, LEfSe is a univariate analysis. The author stated that "Repeating univariate analysis increases the probability of false positives." Can this be mitigated by performaing multi-testing correction, using such as the "BH"- or "BY"-method, which has been widely used in differential gene expression analysis using RNA-seq. How do you reached the conclusion that "LEfSe cannot discover something scientifically meaningful".

5. Lines 114-115, "detects significant differences as significant"? rephrase it?

6. Lines 156 and 157, the former sentence mentioned three patterns, but the latter mentioned 9 patterns. Please clarify.

7. Lines 175-176,  the authors stated that "Our findings will become a Novel Strategies in the Study of the Human Gut Microbiota". It is not clear what their findings are. Please clarify.

8. "the total number of absolute bacterial abundances"? can you say this in a different way?

9. In my opinion, the changes in both the relative abundance and the absolute abundance of mircorbes are importante in some cases. What's the best suggestion to handle this issue?

  175

Comments on the Quality of English Language

See the report.